# Blood Bacteria-Free DNA in Septic Mice Enhances LPS-Induced Inflammation in Mice through Macrophage Response

**DOI:** 10.3390/ijms23031907

**Published:** 2022-02-08

**Authors:** Warerat Kaewduangduen, Peerapat Visitchanakun, Wilasinee Saisorn, Ariya Phawadee, Charintorn Manonitnantawat, Chirapas Chutimaskul, Paweena Susantitaphong, Patcharee Ritprajak, Naraporn Somboonna, Thanya Cheibchalard, Dhammika Leshan Wannigama, Patipark Kueanjinda, Asada Leelahavanichkul

**Affiliations:** 1Department of Microbiology, Faculty of Medicine, Chulalongkorn University, Bangkok 10330, Thailand; porpluemw@gmail.com (W.K.); peerapat.visitchanakun@gmail.com (P.V.); wsaisorn@gmail.com (W.S.); phawadeeariya@docchula.com (A.P.); charintorn.mn@docchula.com (C.M.); chirapas@docchula.com (C.C.); leshanwannigama@gmail.com (D.L.W.); 2Center of Excellence on Translational Research in Inflammation and Immunology (CETRII), Department of Microbiology, Chulalongkorn University, Bangkok 10330, Thailand; 3Nephrology Unit, Department of Medicine, Faculty of Medicine, Chulalongkorn University, Bangkok 10330, Thailand; pesancerinus@hotmail.com; 4Research Unit in Integrative Immuno-Microbial Biochemistry and Bioresponsive Nanomaterials, Department of Microbiology, Faculty of Dentistry, Chulalongkorn University, Bangkok 10330, Thailand; patcharee.r@chula.ac.th; 5Microbiome Research Unit for Probiotics in Food and Cosmetics, Department of Microbiology, Faculty of Science, Chulalongkorn University, Bangkok 10330, Thailand; Naraporn.S@chula.ac.th; 6Program in Biotechnology, Faculty of Science, Chulalongkorn University, Bangkok 10330, Thailand; thanya-_-@hotmail.com; 7Antimicrobial Resistance and Stewardship Research Unit, Department of Microbiology, Faculty of Medicine, Chulalongkorn University, Bangkok 10330, Thailand; 8School of Medicine, Faculty of Health and Medical Sciences, The University of Western Australia, Nedlands, WA 6009, Australia

**Keywords:** sepsis, bacterial DNA, LPS, blood microbiome analysis, cecal ligation and puncture

## Abstract

Although bacteria-free DNA in blood during systemic infection is mainly derived from bacterial death, translocation of the DNA from the gut into the blood circulation (gut translocation) is also possible. Hence, several mouse models with experiments on macrophages were conducted to explore the sources, influences, and impacts of bacteria-free DNA in sepsis. First, bacteria-free DNA and bacteriome in blood were demonstrated in cecal ligation and puncture (CLP) sepsis mice. Second, administration of bacterial lysate (a source of bacterial DNA) in dextran sulfate solution (DSS)-induced mucositis mice elevated blood bacteria-free DNA without bacteremia supported gut translocation of free DNA. The absence of blood bacteria-free DNA in DSS mice without bacterial lysate implies an impact of the abundance of bacterial DNA in intestinal contents on the translocation of free DNA. Third, higher serum cytokines in mice after injection of combined bacterial DNA with lipopolysaccharide (LPS), when compared to LPS injection alone, supported an influence of blood bacteria-free DNA on systemic inflammation. The synergistic effects of free DNA and LPS on macrophage pro-inflammatory responses, as indicated by supernatant cytokines (TNF-α, IL-6, and IL-10), pro-inflammatory genes (*NFκB*, *iNOS*, and *IL-1β*), and profound energy alteration (enhanced glycolysis with reduced mitochondrial functions), which was neutralized by TLR-9 inhibition (chloroquine), were demonstrated. In conclusion, the presence of bacteria-free DNA in sepsis mice is partly due to gut translocation of bacteria-free DNA into the systemic circulation, which would enhance sepsis severity. Inhibition of the responses against bacterial DNA by TLR-9 inhibition could attenuate LPS-DNA synergy in macrophages and might help improve sepsis hyper-inflammation in some situations.

## 1. Introduction

Sepsis, a life-threatening organ failure caused by systemic infection, is a major health care issue worldwide [1]. During sepsis, immune responses against pathogen-associated molecular patterns (PAMPs) from organisms create abnormally uncontrollable inflammation [2], causing multiple organ failure and death [3]. The robust reactions of innate immune cells such as macrophages are the key mechanism of sepsis hyper-inflammation [4] and can be found in almost all organs [5]. Additionally, the immune responses against the circulating free DNA (host and bacteria-free DNA) might be important in sepsis as free DNA is a biomarker to predict sepsis mortality [6,7]. During sepsis, the presence of bacteria-free DNA in serum [6,7] is caused by (i) the breakdown of blood bacteria (bacteremia) [8,9] that originate from the sources of infection and gut translocation [10,11], and (ii) the transfer of bacteria-free DNA from the gut. As such, gut translocation of bacteria-free DNA in patients with intestinal barrier defect without bacteremia (the elderly subjects and patients with severe COVID19) is demonstrated by positive blood bacterial DNA (DNAemia) [12,13] and blood bacterial genome (blood microbiome) [14]. Spontaneous breakdown of bacterial DNA results in small fragments of DNA (65 kDa) [15,16,17,18], a similar size of lipopolysaccharides (LPSs) (50–100 kDa) [19], a Gram-negative bacterial molecule currently used as an indirect biomarker of gut-barrier defect [20,21]. The fragments of free DNA might be small enough to pass through the gut-barrier during sepsis. Although the translocation of bacteria-free DNA from the intestinal bacteria in gut contents might be possible, data on gut translocation of free DNA is still limited.

Furthermore, the influence of bacterial DNA on sepsis severity is still controversial [7,22,23]. Bacteria-free DNA is derived from bacterial death [24,25], especially by bactericidal antibiotics [26,27], worsen sepsis severity [28], and the reduction in bacteria-free DNA by extracorporeal blood purification [29] improves sepsis outcomes [30]. However, serum bacteria-free DNA in patients with sepsis recovery [26,31] or non-bacterial sepsis [32] have no clinical significance. This is different from the solid conclusions on other sepsis worsening factors including bacteremia, tissue hypoxia, cell death, and excessive blood pro-inflammatory cytokines [33], partly because of the lower pro-inflammatory effect of bacterial DNA and unmethylated cytosine-phosphate guanine (CpG) DNA compared with LPS [34,35,36]. Although the synergy between CpG DNA and LPS in mice has been demonstrated [34,35,36], the combination between bacteria-free DNA and LPS might be a better model to resemble sepsis in patients due to some differences between CpG DNA and bacteria-free DNA [37]. For example, differences in the length of DNA and the abundance of the unmethylated CpG motifs between the bacteria-free DNA (directly isolated from bacteria) versus the commercially available CpG DNA might result in varied responses [37]. Regarding the free DNA responses, bacterial DNA activates endosomal TLR-9 [38], TLR-4 on the cell membrane [39], and several cytosol receptors [40,41,42]. While bacteria-free DNA is phagocytosed by phagocytic cells (neutrophils, monocytes, and macrophages) with subsequent TLR-9 activation [43], cytosol insertion of the DNA by lipofectamine or electroporation activates cytosol receptors [24,44]. Although TLR-9 presents in several immune cells (NK cells, monocytes, macrophages, dendritic cells, and neutrophils), macrophages are immune cells that have been frequently used as the representative cells in several studies [45,46,47,48]. The injection of CpG DNA in mice induces mild inflammatory responses, partly through TLR-9 activation in macrophages, while the co-administration of CpG DNA and LPS in mice simultaneously stimulates TLR-4 and TLR-9 in macrophages, which induce more potent responses [34,35,36]. The induction by bacterial DNA might also alter cell energy status in macrophages; however, it needs more proven evidence.

Despite the previous reports on (i) the presence of blood bacteria-free DNA in sepsis [25] and non-sepsis (elderly subjects) [13], and (ii) the enhanced responses against co-administration of synthetic CpG DNA and LPS in mice [34] and macrophages [49,50], the data on gut translocation of bacteria-free DNA, simultaneous analysis of bacteriome from blood and feces, and impact of bacteria-free DNA on cell energy status in macrophages are still very few in number. Hence, several mouse models were conducted for different purposes including (i) a cecal ligation and puncture (CLP) sepsis model, a representative model of sepsis that mostly resemble patients [51], to explore the presence of blood bacteria-free DNA and the simultaneous analysis of bacteriome from blood and feces during sepsis; (ii) a dextran sulfate solution (DSS)-induced gut-barrier defect model with oral gavage of bacterial lysate (a source of bacterial DNA) to examine gut translocation of the free DNA; and (iii) a model with co-administration of bacteria-free DNA and LPS to test the impact of the free DNA against sepsis-inflammatory responses. In addition, the cell energy status, along with several cell characteristics of macrophages after activation by LPS with and without bacteria-free DNA, was also investigated.

## 2. Results

### 2.1. Bacteria-Free DNA in the Blood of Mice with Cecal Ligation and Puncture Sepsis Was Possibly Derived from the Breakdown of Blood Bacteria and Gut Translocation

To demonstrate blood bacteria-free DNA in sepsis, the CLP model, a mouse model that resembles human sepsis [51], was performed. The severity of the CLP model was determined by survival analysis, blood bacterial burdens, serum endotoxin, serum bacteria-free DNA, and gut permeability defect (FITC dextran assay) (Figure 1A–E). The polymicrobial characteristics of CLP sepsis were demonstrated through the mixture of different colony morphologies in blood agar plates that could be identified as Gram-negative bacteria (*E. coli*, *Enterobacter* spp., and *Pseudomonas* spp.), and Gram-positive organisms (*Streptococcus* spp., *Staphylococcus* spp., and *Enterococcus* spp.) (Figure 1F). Sepsis severity was also indicated by serum cytokines, kidney injury, liver damage, and leukopenia (neutropenia and lymphopenia) (Figure 1G–I). Due to the limitation of bacterial identification by morphologies of bacterial colonies from blood culture and the possible gut-derived bacteremia from sepsis-induced gut permeability defect [20,21], the analyses of bacterial microbiome from blood and feces in CLP mice were examined (Figure 2A–E).

Although there was no difference in blood microbiome in the analysis on phylum level between the sham and CLP sepsis mice, the higher abundance of *E. coli* and *Streptococcal* spp. with the lower abundance of Enterobacteriaceae were demonstrated in the analysis on the genus level (Figure 2A,B,E). In the bacteriome analysis, the high abundance of *E. coli* and *Streptococcal* spp. was correlated with the bacterial identification through colony characteristics from blood culture (Figure 1F). In the fecal microbiome analysis (phylum level), CLP enhanced Bacteroides (Gram-negative anaerobe) and Proteobacteria (pathogenic Gram-negative aerobes including *E. coli*), while it reduced Firmicutes (mostly Gram-positive anaerobe) compared with the sham (Figure 2C–E). In the genus level of the fecal microbiome, the analysis was non-different between sham and CLP, partly because of the difference in genus of the bacteria, despite being in the same phylum of the individual CLP mouse (Figure 2C–E). As the predominant Proteobacteria and *E. coli* in the fecal and blood bacteriome, respectively (Figure 2E), some bacteria-free DNAs in serum might be a result of gut translocation of the free DNA. Thus, bacterial DNA in the blood of CLP mice was possibly due to the combination of the breakdown of blood bacteria together with gut translocation.

### 2.2. Bacteria-Free DNA in the Blood of DSS Mice with Oral Gavage by Bacterial Lysate Was Transferred from the Gut Contents

Due to technical limitations on the demonstration of the transfers of bacteria-free DNA from the gut into systemic circulation in the CLP model, a DSS-induced gut-barrier defect model with and without oral gavage by bacterial lysate (a source of bacterial DNA) was used. To test gut translocation without the influence of the breakdown of blood bacteria, gut-barrier damage by 3% DSS was performed with the determination of several parameters (Figure 3A–F). All DSS mice demonstrated acute diarrhea on days 6–7 post-DSS without bacteremia (data not shown), but bacteria-free DNA was detected only in the mice with bacterial lysate oral administration (a source of bacterial DNA) (Figure 3A). The intestinal damage by DSS between mice with or without bacterial lysate was not different as indicated by gut leakage (FITC-dextran assay) (Figure 3B) and colon histology (Figure 3F,G). Meanwhile, endotoxemia and serum inflammatory cytokines were more severe in the DSS mice with bacterial lysate when compared with the DSS alone (Figure 3C–E). With the demonstration of serum bacteria-free DNA in DSS mice, the presence of bacteria-free DNA in CLP sepsis mice might be a result of both blood-bacterial breakdown and gut translocation of the free DNA.

### 2.3. Additive Inflammatory Effect of Bacteria-Free DNA on LPS Responses in LPS Injection Mouse Model

Due to the controversial issues on (i) the impact of blood bacteria-free DNA in sepsis [7,22,23]; (ii) the effects of bacteria-free DNA and LPS co-administration, despite the previous reports using LPS with CpG DNA [34,35,36] that might be different from the isolated bacterial DNA [37]; and (iii) LPS induces gut [20,21] and hepatic injury [52], a mouse model with the co-administration of bacteria-free DNA and LPS was performed. With LPS injection alone, there was an increase in serum bacteria-free DNA (Figure 4A) without bacteremia (data not shown), possibly due to LPS-induced gut leakage (FITC-dextran assay), which facilitated LPS effects, either on hepatic damage (serum alanine transaminase and cytokines from liver tissue) or systemic inflammation (serum cytokines) (Figure 4B–I). At 0.5 h after injection, serum bacteria-free DNA after LPS injection alone was lower than that in mice with the administration of DNA alone and with LPS + DNA co-administration. At 3 h after LPS + DNA injection, the serum bacteria-free DNA was highest among all groups (Figure 4A), implying an additive effect of bacteria-free DNA from LPS-induced gut-barrier defect and direct bacteria-free DNA injection. Administration of bacteria-free DNA alone caused lower gut-barrier defect, hepatic injury (liver enzyme and cytokines in liver tissue), and serum cytokines when compared to LPS + DNA in some time-points after the injection (Figure 4B–I). In comparison with LPS injection alone, LPS + DNA induced higher bacteria-free DNA, liver enzyme, serum cytokines, and liver IL-6 in some time-points after the injection (Figure 4A–I). Hence, the presence of bacteria-free DNA may increase the pro-inflammatory effects during sepsis, partly through the synergy with endotoxemia.

### 2.4. Additive Inflammatory Effect of Bacteria-Free DNA on LPS Responses, a Possible Impact of Cell Energy Alteration in Macrophages

Due to the role of macrophages in recognizing pathogen molecules (both LPS and bacteria-free DNA) [49] and the lack of data on macrophage energy after free DNA activation, in vitro experiments on bone marrow-derived macrophages were performed. Accordingly, LPS + DNA induced more prominent inflammation than LPS alone, as indicated by an increase in supernatant cytokines (TNF-α, IL-6, and IL-10), gene expression of pro-inflammatory signals (*NFκB*, *iNOS,* and *IL-1β* but not *TLR-4*), and anti-inflammatory signals (*Arg-1* but not *TGF-β* and *Fizz-1*) (Figure 5A–J). Despite the association with the potent intensity of TLR-4 activation [53], the downregulation of *TLR-4* was not different between the LPS and LPS + DNA groups (Figure 5E). With bacteria-free DNA activation alone, supernatant IL-10, but not TNF-α and IL-6, was higher than in the control media but was lower with LPS activation alone (Figure 5A–C). The *NFκB* expression in DNA-activated macrophages was higher than in the LPS-activation alone and lower than in the LPS + DNA activated macrophages (Figure 5D–J). The activation by bacteria-free DNA alone upregulated *TLR-4* expression at the highest level without an alteration in genes for macrophage polarization (Figure 5D–J). These data support mild macrophage immune activation by bacteria-free DNA alone and further pro-inflammatory in DNA + LPS co-stimulated macrophages [34,35,36].

All immune activations were shown to affect cell energy status by reducing the expression of mitochondrial DNA (*mtDNA*) and mitochondrial abundance (fluorescent staining) (Figure 6A,B). The mtDNA was lowest in LPS + DNA activated macrophages (Figure 6A,B), which supports the profound energy depletion and cell injury after potent immune responses [54]. Likewise, there was a prominent reduction in respiratory capacity and respiratory reserve in LPS + DNA activated macrophages, while LPS or DNA activation alone did not alter the mitochondrial stress test (Figure 6C–E). Although all activations similarly enhanced glycolysis capacity and glycolysis reserve, the glycolysis reserve in LPS + DNA was higher than the other groups (Figure 6F–H). Due to the dominant energy sources from the glycolysis pathway during cytokine production [55,56,57], an increase in glycolysis reserve in LPS + DNA supports the pro-inflammatory effects of bacterial DNA on the cell energy status of LPS-activated macrophages.

Because TLR-9 is a downstream signal of bacterial DNA activation, which requires proper endosomal acidification for proper activities [58], the inhibitors against endosomal acidification (chloroquine and monensin) were used as representative TLR-9 interferences. Both inhibitors attenuated responses against LPS + DNA, as indicated by supernatant cytokines (TNF-α, IL-6, and IL-10) and *NFκB* expression (Figure 7A–D). Despite mild macrophage responses against DNA stimulation alone, the inhibitors reduced the supernatant IL-10 and gene expression of *NFκB* and *TLR-4* (Figure 7C–E), supporting the recognition of bacteria-free DNA by TLR-9 [59,60]. Due to (i) high glycolysis activity in macrophages facilitating cytokine production [56] and (ii) chloroquine decreasing macrophage cytokine production [61] and altering energy status in some cells [62], the reduction in LPS-induced macrophage cytokine production by chloroquine (Figure 7A–C) might be a result of interference on cell energy status. In LPS + DNA stimulated macrophages, chloroquine lowered glycolysis capacity as well as mitochondrial respiratory capacity, but not glycolysis reserve nor respiratory reserve (Figure 8A–C). Although the reduced energy status of LPS + DNA activated macrophages by chloroquine might cause a decrease in cytokine production, the anti-inflammatory effect of chloroquine was insufficient to protect mitochondrial injury, as indicated by the non-difference in *mtDNA* and mitochondrial abundance (fluorescent staining) between LPS + DNA with and without chloroquine (Figure 8G,H).

## 3. Discussion

Although bacteria-free DNA is a serum pathogen molecule with a potential pro-inflammatory property, there is debate on its role in immune responses [7,22,23] and the adjunctive interventions to reduce blood-free DNA during hyper-inflammatory sepsis [30]. The presence of bacteria-free DNA induces significant stress in patients with sepsis as microbial DNA induces more potent inflammatory activity than the host DNA [34]. The sources of bacteria-free DNA in blood during sepsis were possibly not only a result of the breakdown of bacteria in the blood, but also from the gut translocation due to a sepsis-induced gut-barrier defect [20,21]. Although the intact bacterial DNAs (genome) at a molecular size of 100 to 15,000-kilo base-pair (kbp) (6.5 × 10^4^–9.8 × 10^6^ kDa [63,64]) cannot pass through the gut-barrier, the bacteria-free DNA is rapidly naturally broken down by several processes (depurination or deamination DNA) [65,66] into smaller sizes of less than 100 bp (65 kDa) [15,16,17,18], which can pass through the gut-barrier in several models [65,66].

Here, several mouse models were used to test the hypotheses sequentially (Figure 9). First, the CLP model, a sepsis model that more resembles patients than the LPS injection model [51], was used to explore the presence of bacteria-free DNA in blood. As such, CLP surgery induces sepsis with bacteremia (a common sepsis characteristic in the patients), while LPS injection activates hyper-inflammation without bacteremia, referred to as “a chemical toxin injection model” [67,68]. Blood bacteria-free DNA was detected in CLP mice as indicated by real-time PCR and blood bacteriome analysis. With limited data on the simultaneous analysis of bacteriome in feces and blood, we demonstrated a prominent abundance of Proteobacteria (phylum) (a group of pathogenic bacteria) in both the sham and CLP mice. An abundance of *E. coli*, a Proteobacteria that commonly cause gut-derived sepsis [21], was higher in CLP sepsis than sham mice. Although Firmicutes (phylum) was the most prominent bacteria in feces in both sham and CLP mice, the abundance of Firmicutes (a group of the possible beneficial bacteria [69]) in blood bacteriome was lower than Proteobacteria. These findings indicate the greater invasiveness of Proteobacteria than Firmicutes [70]. Moreover, Proteobacteria in the blood of sham mice also implies a possible transient gut translocation in healthy mice as normal physiology, especially through the induction with pathogenic bacteria (such as Proteobacteria), despite an intact gut barrier by the FITC-dextran test [71,72]. With the CLP model, we proposed that bacteria-free DNA in sepsis serum was possibly from not only the bacterial breakdown but also the gut translocation (Figure 9), which partly correlated with the invasiveness of Proteobacteria.

Second, gut translocation of bacteria-free DNA was highlighted in DSS mice with oral gavage of bacterial lysate (a source of bacterial DNA and LPS). An increase in microbe molecules in gut contents by the oral gavage of bacterial lysate enhanced the detection of bacteria-free DNA (and increased endotoxemia) in the serum. Unlike the CLP model, bacteria-free DNA in the serum of DSS mice results from the gut translocation, but not the bacterial breakdown due to an absence of bacteremia in the DSS model. As the bacterial gavage did not enhance the severity of DSS-induced mucositis (pathological analysis and FITC-dextran assay), an increase in LPS and free DNA in the serum of the lysate-administered DSS mice were from gut translocation, and not the enhanced severity of mucositis. Because of the similar molecular size between LPS and fragments of bacteria-free DNA [15,16,17,18], an increase in LPS and bacteria-free DNA in gut contents by bacterial lysate administration caused an increase in endotoxemia and blood bacteria-free DNA. Hence, we propose that an adequate amount of bacteria-free DNA in gut content along with gut permeability defect can induce gut translocation of bacteria-free DNA, as demonstrated by the bacterial lysate-administered DSS mice (Figure 9).

Third, the injection of LPS and bacteria-free DNA model was used to explore the influence of bacteria-free DNA in sepsis responses. Although LPS injection has a limitation on the resemblance to sepsis in patients [67,68], direct injection is frequently used to test immune responses. Despite previous publications on CpG DNA-LPS co-administration in mice [34] and the differences between bacteria-free DNA and CpG DNA [37], the data on the utilization of isolated bacteria-free DNA are limited. As such, the injection of LPS induced serum bacterial-free DNA, but did not cause bacteremia in mice [56,73], supporting LPS-induced bacteria-free DNA translocation from the gut [74]. In parallel, LPS injection plus bacterial DNA (LPS + DNA) induced more severe hepatic injury (liver enzyme and IL-6) than LPS alone (Figure 9), perhaps due to the impact of microbial DNA on immune responses. The co-existence of LPS and bacterial DNA in the serum of sepsis mice (CLP and LPS models) may enhance systemic inflammation through macrophage M1 polarization (*iNOS* and *IL-1β* expression) [34,35] and *NFκB* upregulation. Indeed, in macrophages with DNA stimulation alone, there was less potent pro-inflammatory activation (low supernatant IL-10) with *TLR-4* upregulation [75], while LPS + DNA exhibited a profound inflammatory response (Figure 9). Notably, the similar downregulation of *TLR-4*, perhaps from the TLR-4 internalization (endocytosis) [53] between LPS and LPS + DNA activation suggests that bacteria-free DNA did not activate TLR-4, despite a possible DNA-induced TLR-4 activation in some situations [39].

From all animal models, we demonstrated the gut translocation of bacteria-free DNA in sepsis with LPS-bacterial DNA synergy, possibly through the simultaneous TLR-4 and TLR-9 activation on macrophages [49,50]. The profound inflammation induces mitochondrial injury [76,77] and mitochondrial free DNA in systemic circulation, which further facilitate inflammatory responses [78,79] in several immune cells including macrophages [80]. Here, LPS induced M1 macrophage polarization [56], partly through the acceleration of glycolysis, the main energy source of the cells during the hyper-inflammatory stage, despite the reduction in mitochondrial activities (mitochondrial abundance, mtDNA, and maximal respiration), supporting previous publications [81,82]. The bacterial DNA reduced macrophage mitochondrial functions, similar to LPS activation. However, the DNA stimulation alone did not enhance macrophage glycolysis, resulting in a lower cytokine production when compared to the LPS activation [56]. The LPS-bacterial DNA synergy might be due to the simultaneous activation of the endosomal TLR-9 [83] and TLR-4 by DNA and LPS activation, respectively [73]. This process requires high energy utilization, which results in mitochondrial injury and further inflammatory activation (Figure 10). Inhibition of endosomal acidification (a TLR-9 interference) neutralized LPS-bacterial DNA synergy and reduced mitochondrial injury (improved respiratory capacity). Because (i) LPS and bacterial DNA are simultaneously presented in sepsis [84] and (ii) inhibition of endosomal acidification (chloroquine) improves sepsis severity in animal models [85], our data support an influence of bacteria-free DNA on enhanced sepsis severity [83]. Despite debate on the use of hydroxychloroquine in sepsis, especially in acute respiratory distress syndrome from Coronavirus (COVID-19) infection [86,87], there might be some sepsis conditions that are beneficial from TLR-9 inhibition and bacterial DNA neutralization [88]. Several extracorporeal blood purification procedures reduce the free DNA particles in either sepsis [29] or non-sepsis (viral infection) [89,90,91], which could be adapted for the removal of bacteria-free DNA in sepsis.

Finally, there were several limitations to our study. First, the mortality rate of the CLP model was very high when compared to other studies, possibly due to the lack of antibiotic use [95,96,97,98,99,100]. Second, the gut translocation of bacterial DNA was indirectly determined by the measurement in serum, but not the direct intestinal imaging. Third, the mechanisms of sepsis in the animal models and the in vitro experiments might have some differences from the patients [101,102]. Finally, TLR-9 is also expressed in other immune cells such as dendritic cells and NK cells, which might also affect immunologic responses against bacteria-free DNA [103,104,105]. More studies on this topic are warranted.

In conclusion, gut translocation of bacterial DNA enhanced the synergistic activation of LPS and bacteria-free DNA on inflammatory responses and mitochondrial dysfunction. Inhibition of DNA activation might be an interesting strategy to improve sepsis outcomes in some situations and should be further tested.

## 4. Materials and Methods

### 4.1. Animal

The animal study protocol (025/2563) was endorsed by the Institutional Animal Care and Use Committee of Faculty of Medicine, Chulalongkorn University following the U.S. National Institutes of Health (NIH) animal care and use protocol. Wild-type C57BL/6J (WT) mice were purchased from Nomura Siam (Pathumwan, Bangkok, Thailand) and only male 8-wk-old mice weighing approximately 20–22 g were used in the experiments. The mice were housed in standard clear plastic cages (3–5 mice per cage) with free access to water and food (SmartHeart Rodent; Perfect companion pet care, Bangkok, Thailand) and a light/dark cycle of 12:12 h in 22 ± 2 °C with 50 ± 10% relative humidity.

### 4.2. Cecal Ligation and Puncture Model and Lipopolysaccharide with and without Bacterial DNA Injection Model

Sepsis was induced by cecal ligation and puncture (CLP) surgery or lipopolysaccharide (LPS) injection with or without bacterial DNA. As such, CLP was conducted under isoflurane anesthesia following a previously published protocol [106]. Briefly, the cecum was ligated and punctured twice with a 21-gauge needle in CLP, while the sham was only an operation for cecum identification. Mice were sacrificed via cardiac puncture under isoflurane anesthesia with blood collection at 24- and 96-h post-operation for acute sepsis parameters and for survival analysis, respectively. Serum was kept at −80 °C until analysis. To test the impact of bacterial DNA on sepsis mice, bacterial DNA was administered with and without LPS in mice. Accordingly, DNA was extracted from *Escherichia coli* (ATCC 25922) (ATCC, Manassas, VA, USA) following a published protocol [75]. Briefly, *E. coli* in Luria–Bertani (LB) broth isolated DNA using the Tissue Genomic DNA Extraction Mini Kit (Favorgen Biotech, Wembley, WA, Australia) and quantified by a Nanodrop ND-100 (Thermo Scientific, Waltham, MA, USA). Then, mice were intraperitoneal (ip) administered with 10 mg/kg of LPS from *E. coli* 026: B6 (Sigma-Aldrich, St. Louis, MO, USA) with or without intravenous (iv) administration of the extracted bacterial DNA at 10 mg/kg (or DNA alone). Blood samples were collected through the tail vein at three days before LPS-injection (0 h) and 0.5 h post-administration. Then, mice were sacrificed at 3 h post-injection by cardiac puncture under isoflurane anesthesia with sample collection. Serum was kept at −80 °C until use and the liver was snap-frozen in liquid nitrogen for further analysis. A total 32 and 26 mice were used for CLP and the injection of the pathogen molecule experiments, respectively.

### 4.3. Dextran Sulfate Solution Model with Bacterial Lysate Administration

Gut permeability defect (gut leakage) induced by dextran sulfate solution (DSS) with or without bacterial lysate administration was performed to explore a possible gut translocation of bacterial molecules. Accordingly, DSS (Sigma-Aldrich, St. Louis, MO, USA) was prepared by dilution into drinking water at the concentration of 3% volume by volume (*v/v*) for one week [107]. Bacterial lysate of *E. coli* (ATCC 25922) using *E. coli* at 4.8 × 10^8^ CFU in 0.3 mL phosphate buffer solution (PBS) was orally administered once-daily for three consecutive days starting from the fifth day of the DSS model. At 6 h after the last dose of bacterial lysate on the seventh day of DSS, mice were sacrificed by cardiac puncture under isoflurane anesthesia with sample collection. Serum was kept at −80 °C until use and the organ was put in 10% neutral formalin for histological analysis. All 28 mice were used for this experiment.

### 4.4. Blood Sample Analysis and Gut Permeability Measurement

Serum endotoxin (LPS) was measured by the Limulus Amebocyte lysate test (Associates of Cape Cod, East Falmouth, MA, USA), and values of LPS < 0.01 EU/mL were recorded as 0 due to the limitation of the standard curve. Kidney injury (serum creatinine) and liver damage (alanine transaminase; ALT) were determined by QuantiChrom Creatinine-Assay (DICT-500) and EnzyChrom ALT assay (EALT-100) (BioAssay, Hayward, CA, USA). Serum cytokines were measured by the enzyme-linked immunosorbent assay (ELISA) (Invitrogen, Waltham, MA, USA). For peripheral blood leukocytes, blood was mixed with 3% *v/v* of acetic acid for red blood cell lysis in a ratio of blood and acetic acid at 1:20 by volume before counting with a hemocytometer. Then, a Wright-stained blood smear was determined for the percentage of neutrophils and lymphocytes. The total number of these cells was calculated by the total count from the hemocytometer multiplied by the percentage of cells from the Wright-stained slide. Blood in serial dilutions was directly spread onto blood agar (Oxoid, Hampshire, UK) and incubated at 37 °C for 24 h before colony enumeration. The colonies were identified by mass spectrometry analysis (Vitek MS; bioMérieux SA, Marcy-l’Etoile, France), according to the routine hospital protocol.

To determine bacterial-free DNA, the DNA in serum was extracted with 5 M potassium acetate/acetic acid buffer and quantified by a Nanodrop 100 spectrophotometer (Thermo Scientific, Waltham, MA, USA). Afterward, the samples were examined using the 16s primer 5′-GATGAACGCTGGCGGCGTGC-3′ (F), 5′-CAATCATTTGTCCCACCTTC-3′ (R) by quantitative real-time polymerase chain reaction (PCR) with QuantStudio 6 Flex Real-time PCR System to determine bacterial DNA from the cell-free DNA preparations. For gut permeability determination, FITC-dextran, a 4.4 kDa intestinal nonabsorbable molecule (Sigma-Aldrich, St. Louis, MO, USA), at 12.5 mg per 25 g mouse was orally administered at 3 h before the detection of FITC-dextran in serum using a fluorospectrometer (NanoDrop 3300; ThermoFisher Scientific, Wilmington, DE, USA). The presence of non-intestinal absorbable molecules in serum after oral administration indicates gut permeability defect (gut leakage). Due to the possibility of liver injury during the detoxification of foreign molecules in the blood [108], cytokines from liver tissue after the activation by LPS and/or bacterial DNA were evaluated following a previously published protocol [109]. Briefly, liver samples were weighed, sonicated thoroughly, and the supernatant from homogenous tissue preparation was collected for cytokine measurement by the ELISA assay (Invitrogen, Waltham, MA, USA).

### 4.5. Histology

The semi-quantitative evaluation of intestinal histology on hematoxylin and eosin (H&E) staining at 200× magnification was determined [107] based on mononuclear cell infiltration (in mucosa and sub-mucosa), epithelial hyperplasia (epithelial cell in longitudinal crypts), reduction in goblet cells, and epithelial vacuolization in comparison with control groups using the following scores; 0: leukocyte < 5% and no epithelial hyperplasia (<10% of control); 1: leukocyte infiltration 5–10% or hyperplasia 10–25%; 2: leukocyte infiltration 10–25% or hyperplasia 25–50% or reduced goblet cells (>25% of control); 3: leukocyte infiltration 25–50% or hyperplasia > 50% or intestinal vacuolization; 4: leukocyte infiltration > 50% or ulceration.

### 4.6. Microbiome Analysis of Feces and Blood

Feces (0.25 g) and blood (200 µL) from each mouse were collected from the mice in different cages to avoid the influence of allocoprophagy on the microbiome analyses. Then, the fecal and blood microbiota analyses were performed as previously described [110]. Briefly, the metagenomic DNA was extracted from the prepared samples using a DNAeasy Kit (for feces) and QIAamp (for blood) (Qiagen, Redwood City, CA, USA) with DNA quality assessment using a Nanodrop spectrophotometer. Universal prokaryotic primers 515F (5′-GTGCCAGCMGCCGCGGTAA-3′) and 806R (5′-GGACTACHVGGGTWTCTAAT-3′) with appended 50 Illumina adapter and 30 Golay barcode sequences were used for 16S rRNA gene V4 library construction. The bioinformatic analyses were performed following Mothur’s Standard Operating Procedures (SOP) [111].

### 4.7. Macrophage Experiments and Inhibition of Endosomal Acidification

Macrophages were derived from the bone marrow of the healthy mice as previously described [55]. Briefly, bone marrow from femurs and tibias were collected by centrifugation at 6000 rpm for 4 °C and incubated for seven days with Dulbecco’s modified Eagle medium (DMEM) supplemented with 10% fetal bovine serum (FBS), 1% penicillin/streptomycin, 4-(2-hydroxyethyl)-1-piperazineethanesulfonic acid (HEPES) with sodium pyruvate in a humidified 5% CO_2_ incubator at 37 °C. Conditioned media of the L929 cell line, containing macrophage-colony stimulating factor, at 20% weight by volume (w/v) were used to induce macrophages from the pluripotent stem cells. Then, *E. coli* DNA from the extraction (previously mentioned) at 5 ng/µL with or without LPS (*E. coli* 026: B6, Sigma-Aldrich) at 100 ng/mL) or media control alone (DMEM) were incubated with macrophages at 1 × 10^5^ cells/well at 37 °C for 24 h before the determination of supernatant cytokines (TNF-α, IL-6, and IL-10) using the ELISA assay (Invitrogen). Total RNA was prepared by Trizol, quantified by a Nanodrop ND-1000 (Thermo Fisher Scientific), converted into cDNA by the Reverse Transcription System, and performed real-time quantitative reverse transcription-polymerase chain reaction (qRT-PCR) using the SYBR Green system (Applied biosystem, Foster City, CA, USA) for the expression of several genes [55]. The cDNA template and target primers based on the ΔΔCT method (2-∆∆Ct) with the *β-actin* housekeeping gene were used. Primers for inflammatory signals (*TLR-4* and *NF-κB*), M1 pro-inflammatory macrophage polarization (*iNOS* and *IL-1β*), and M2 anti-inflammatory macrophage polarization (*Fizz-1*, *Arginase-1*, and *TGF-β*) were used (Table 1). Because endosomal acidification and maturation are required for proper TLR9 signaling [58], chloroquine (15 µM) or monensin (20 µM) (Sigma-Aldrich) were pre-incubated with macrophages for 2 h before incubation with several stimulators (bacterial DNA with or without LPS as above-mentioned) to investigate the influence of TLR9 [58].

### 4.8. Mitochondrial Evaluation and Extracellular Flux Analysis

Mitochondrial enumeration was evaluated by mitochondrial DNA (mtDNA) quantification using a Tissue Genomic DNA Extraction Mini Kit (Favorgen Biotech, Wembley, WA, Australia) with a Nanodrop ND-100 (Thermo scientific) via the ΔΔCT method using β2-microglobulin (β2M) normalization [55]. In parallel, mitochondrial biogenesis was analyzed by MitoTracker using 200 nM of MitoTracker Red CMxROS (Molecular probe, Eugene, OR, USA), which was incubated at 37 °C for 15 min before fixing with cold methanol at −20 °C and measured by a microplate reader at excitation OD579 nm and emission OD599 nm [55]. Cell energy status was determined by extracellular flux analysis using Seahorse XFp Analyzers (Agilent, Santa Clara, CA, USA) with oxygen consumption rate (OCR) and extracellular acidification rate (ECAR) representing mitochondrial function (respiration) and glycolysis activity, respectively [56]. Briefly, the stimulated macrophages at 1 × 10^5^ cells/well with different conditions were incubated in Seahorse media (DMEM complemented with glucose, pyruvate, and L-glutamine) (Agilent, 103575–100) for 1 h before activation by different metabolic interference compounds including oligomycin, carbonyl cyanide-4-(trifluoromethoxy)-phenylhydrazone (FCCP), and rotenone/antimycin A for OCR evaluation. In parallel, glycolysis stress tests were performed using glucose, oligomycin, and 2-deoxy-d-glucose (2-DG) for ECAR measurement. The data were analyzed by Seahorse Wave 2.6 software based on the following equations: (i) respiratory capacity (maximal respiration) = OCR between FCCP and rotenone/antimycin A − OCR after rotenone/antimycin A; (ii) respiratory reserve = OCR between FCCP and rotenone/antimycin A − OCR before oligomycin; (iii) maximal glycolysis (glycolysis capacity) = ECAR between oligomycin and 2-DG − ECAR after 2-DG; and (iv) glycolysis reserve = ECAR between oligomycin and 2-DG − ECAR between glucose and oligomycin.

### 4.9. Statistical Analysis

All data were analyzed by the Statistical Package for Social Sciences software (SPSS 22.0, SPSS Inc., Chicago, IL, USA) and Graph Pad Prism version 7.0 software (La Jolla, CA, USA). The results were presented as mean ± standard error (SE). Survival analysis was determined by a Log-rank test. The differences between two or multiple groups were examined for statistical significance by the Student’s T-test or one-way analysis of variance (ANOVA) with Tukey’s analysis of comparison, respectively. The time-point experiments were analyzed by repeated measures ANOVA. A *p*-value < 0.05 was considered statistically significant.

## Figures and Tables

**Figure 1 ijms-23-01907-f001:**
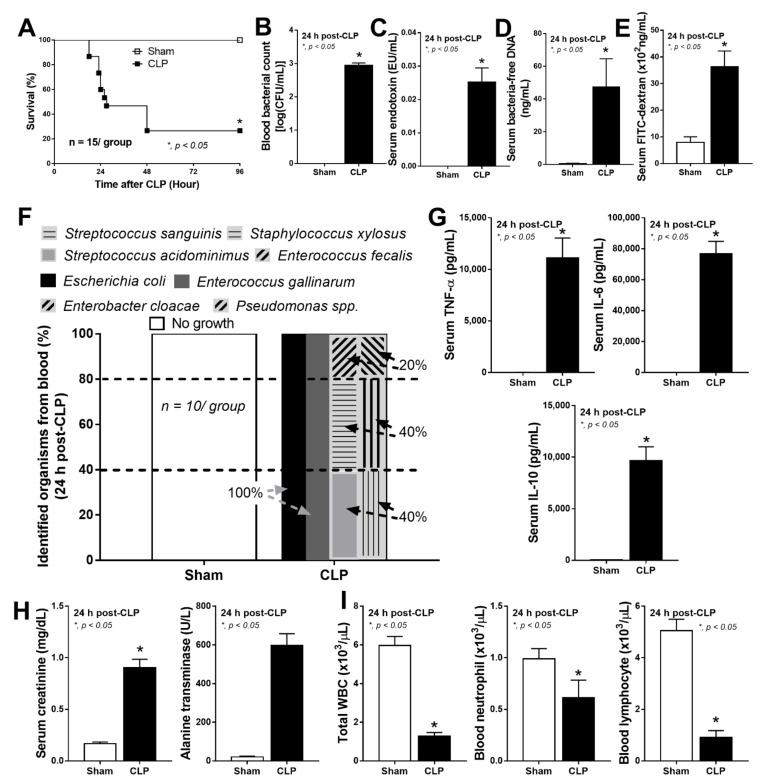
Bacteremia in cecal ligation and puncture (CLP) sepsis, the impact of both intestinal infection and gut-barrier defect. Characteristics of mice after sham or CLP, as indicated by survival analysis (**A**), bacterial burdens in the blood (**B**), serum endotoxin (**C**), serum bacteria-free DNA (**D**), gut-barrier defect (FITC-dextran assay) (**E**), identified bacteria based on bacterial colony characteristics (mass-spectrometry analysis) (**F**), serum cytokines (TNF-α, IL-6, and IL-10) (**G**), organ injury (serum creatinine and alanine transaminase) (**H**), and peripheral blood leukocyte (total, neutrophils, and lymphocytes) (**I**) are demonstrated (*n* = 15/group for A and *n* = 9–10/group for **B**–**I**). *, *p* < 0.05 vs. sham; survival analysis and the difference between two groups were determined by Log-rank test and Student’s T test, respectively.

**Figure 2 ijms-23-01907-f002:**
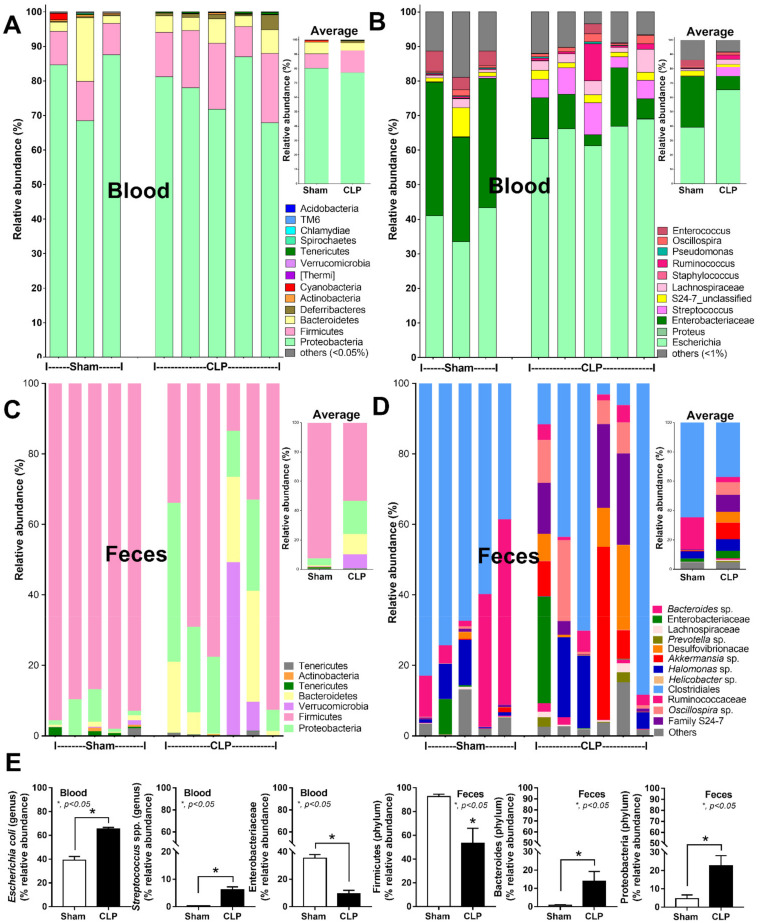
Pathogenic proteobacteria were mainly presented in blood bacteriome of mice, both sham and cecal ligation and puncture (CLP), while Firmicutes were mainly demonstrated in fecal microbiome analysis of both groups. Characteristics of bacteriome analysis from the blood and feces of mice after sham or CLP, as indicated by bacterial abundance in phylum and species with the average abundance (**A**–**D**) and the graph presentation of some groups of bacteria (**E**) are demonstrated (*n* = 3–5 for sham and *n* = 6 for CLP). *, *p* < 0.05 vs. sham; the difference between groups was determined by Log-rank test and Student’s T test, respectively.

**Figure 3 ijms-23-01907-f003:**
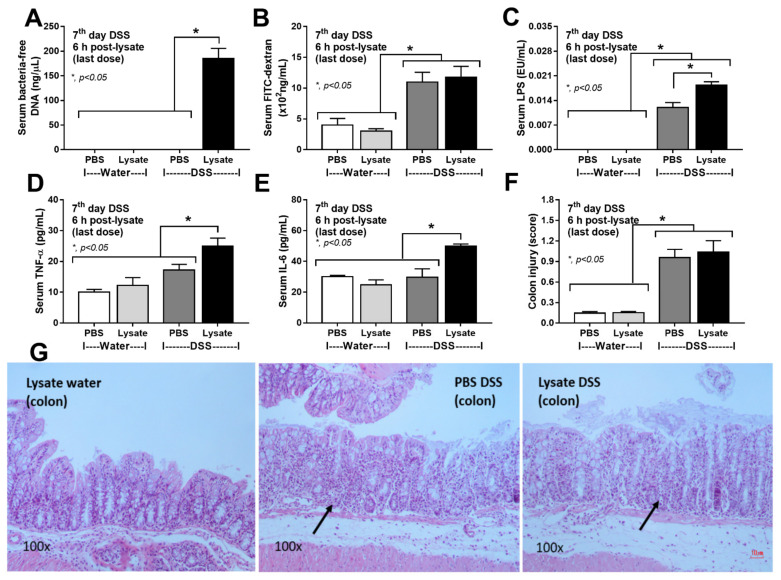
The translocation of bacteria-free DNA from the gut into the blood circulation in a non-surgical intestinal tight-junction injury model using dextran sulfate solution (DSS)-induced mucositis. Characteristics of mice with DSS or regular drinking water (Water) with oral gavage by bacterial lysate or phosphate buffer solution (PBS) started on days 5–7 of the experiments, as indicated by serum bacteria-free DNA (**A**), gut-barrier defect (FITC-dextran) (**B**), serum lipopolysaccharide (LPS) (**C**), serum pro-inflammatory cytokines (TNF-α and IL-6) (**D**,**E**) and colon injury score with representative histological pictures on hematoxylin and eosin (H&E) staining (**F**,**G**) are demonstrated (*n* = 6–8/group). The colon pathology of control PBS gavage with regular drinking water (PBS–water) is not demonstrated due to the similarity with control bacterial lysate gavage with water (lysate–water). Arrows indicate inflammatory cell infiltration in DSS-induced intestinal injury. *, *p* < 0.05 between the indicated groups; the difference between groups was determined by one-way ANOVA with Tukey’s analysis.

**Figure 4 ijms-23-01907-f004:**
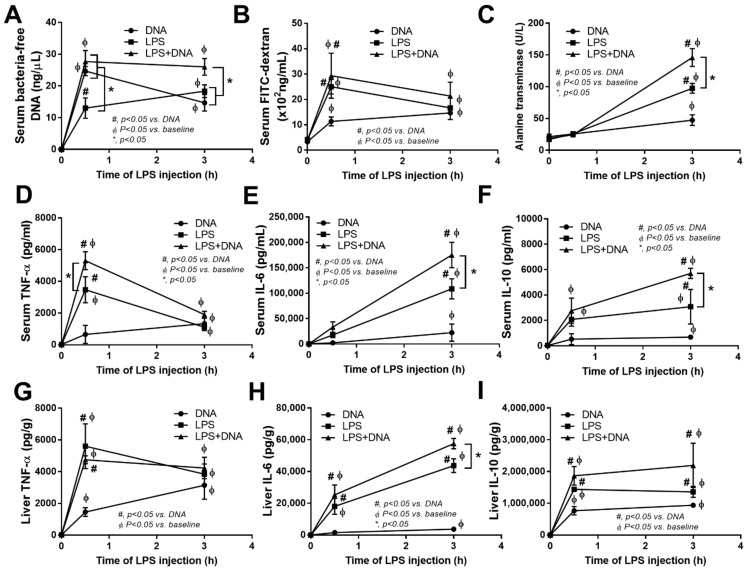
Additive effect of bacteria-free DNA on LPS-induced pro-inflammation in mice and the influence of bacteria-free DNA in blood circulation. Characteristics of mice after injection with bacteria-free DNA (DNA) or lipopolysaccharide (LPS) alone or in combination (LPS + DNA), as indicated by serum bacteria-free DNA (**A**), gut-barrier defect (FITC-dextran) (**B**), serum alanine transaminase (**C**), serum cytokines (TNF-α, IL-6, and IL-10) (**D**–**F**), cytokines from liver tissue (TNF-α, IL-6, and IL-10) (**G**–**I**) are demonstrated (*n* = 7–9/time-point). #, *p* < 0.05 vs. DNA; ϕ, *p* < 0.05 vs. baseline; *, *p* < 0.05 between the indicated groups; one-way ANOVA with Tukey’s analysis and repeated measures ANOVA were used for the analysis among groups and between different time-points within groups, respectively.

**Figure 5 ijms-23-01907-f005:**
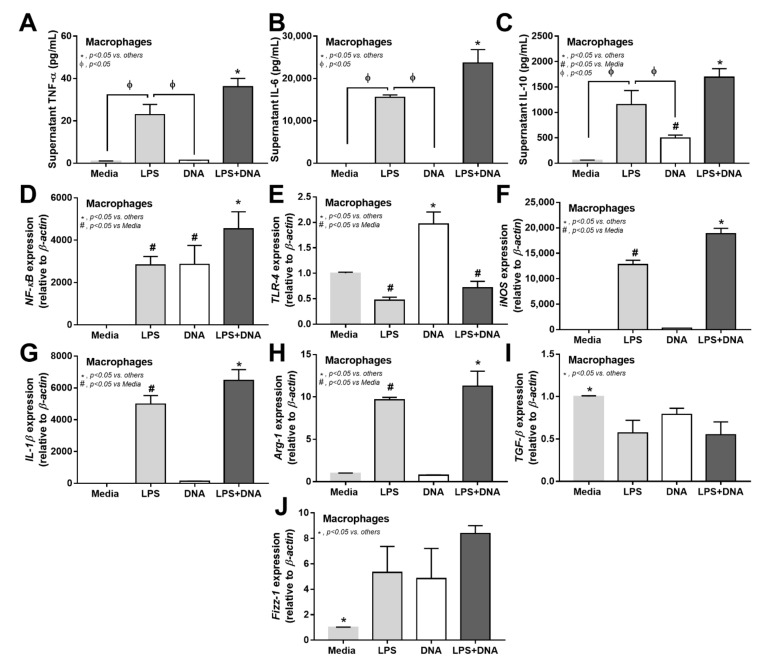
Additive effect of bacteria-free DNA on LPS-induced pro-inflammation M1 macrophage polarization and the impact of bacterial DNA in macrophage responses. Characteristics of macrophages after activation by media control, lipopolysaccharide (LPS), bacteria-free DNA (DNA), and LPS with the DNA (LPS + DNA), as indicated by supernatant cytokines (TNF-α, IL-6, and IL-10) (**A**–**C**) and the expression of genes for inflammatory signals (*NFκB* and *TLR-4*) (**D**,**E**), M1 macrophage polarization (*iNOS* and *IL-1β*) (**F**,**G**), and M2 macrophage polarization (*FIZZ-1*, *Arg-1,* and *TGF-β*) (**H**–**J**) are demonstrated (independent triplicated experiments were performed). *, *p* < 0.05 vs. all of other groups; #, *p* < 0.05 vs. macrophages with media control group; the difference between groups was determined by one-way ANOVA with Tukey’s analysis.

**Figure 6 ijms-23-01907-f006:**
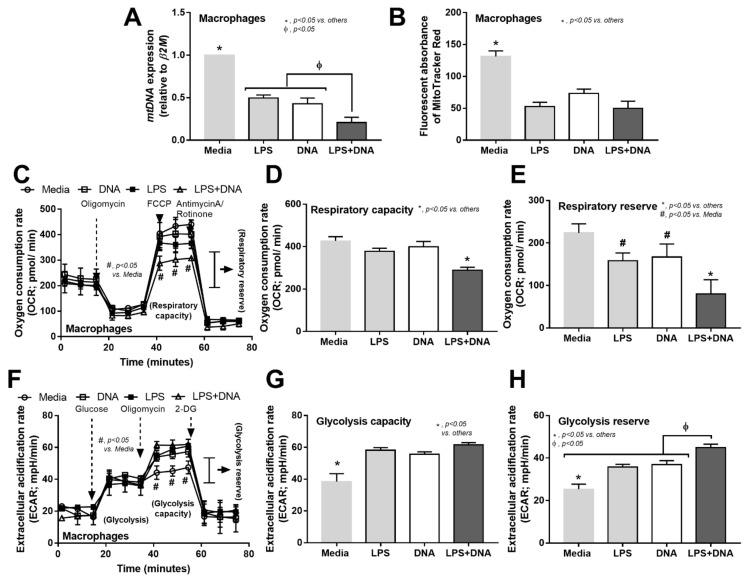
Additive effect of bacteria-free DNA on LPS-mediated cell energy alteration in macrophages and the impact of bacteria-free DNA in macrophage responses. Characteristics of macrophages after activation by media control, lipopolysaccharide (LPS), bacteria-free DNA (DNA), and LPS with the DNA (LPS + DNA), as indicated by mitochondrial abundance using mitochondrial DNA (*mtDNA*) and MitoTracker fluorescent staining (**A**,**B**), oxygen consumption rate (OCR) of mitochondrial stress test with the graph presentation of respiratory capacity (maximal respiration) and respiratory reserve (**C**–**E**), and extracellular acidification rate (ECAR) for a glycolysis stress test with the graph presentation of glycolysis capacity (maximal glycolysis) and glycolysis reserve (**F**–**H**) are demonstrated (independent triplicated experiments were performed). *, *p* < 0.05 vs. all of the other groups; ϕ, *p* < 0.05 between the indicated groups; the difference between groups was determined by one-way ANOVA with Tukey’s analysis.

**Figure 7 ijms-23-01907-f007:**
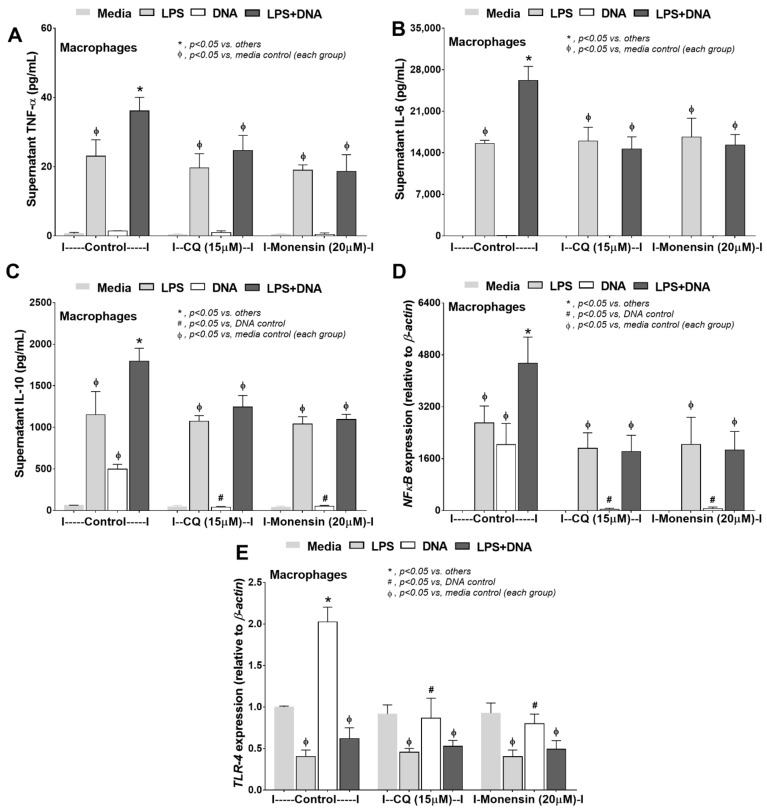
TLR-9 inhibitors attenuated macrophage inflammatory responses against bacteria-free DNA and the impact of TLR-9 on bacteria-free DNA-induced inflammation. Characteristics of macrophages after activation by media control, lipopolysaccharide (LPS), bacteria-free DNA (DNA), and LPS with the DNA (LPS + DNA) with incubation by media (Control) or endosomal acidification inhibitors, chloroquine (CQ) or monensin, as indicated by supernatant cytokines (TNF-α, IL-6, and IL-10) (**A**–**C**) and expression of inflammatory genes (*NFκB* and *TLR-4*) (**D**,**E**) are demonstrated (independent triplicated experiments were performed). *, *p* < 0.05 vs. all of other groups; #, *p* < 0.05 vs. macrophages with DNA in the control group; ϕ, *p* < 0.05 vs. macrophages with media control within each group of the experiments; the difference between groups was determined by one-way ANOVA with Tukey’s analysis.

**Figure 8 ijms-23-01907-f008:**
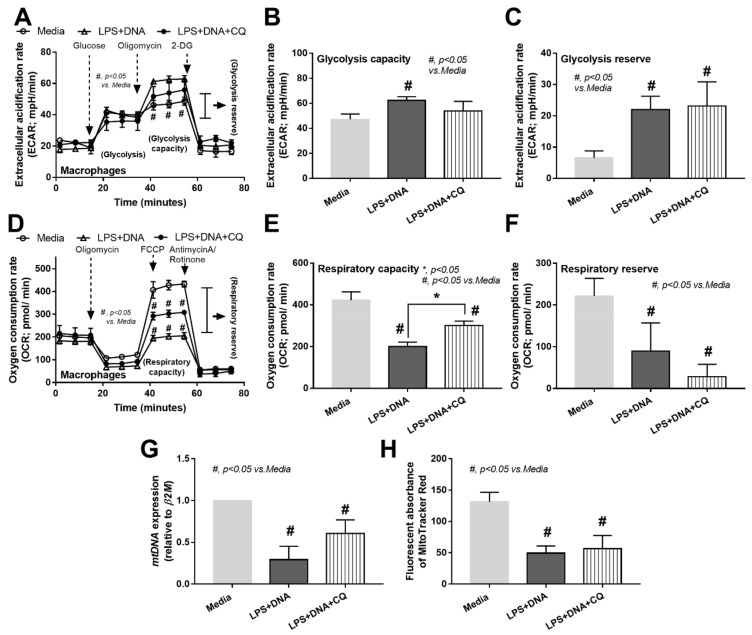
TLR-9 inhibitors attenuated energy modification in macrophage with bacteria-free DNA stimulation and the impact of TLR-9 on bacteria-free DNA-induced cell-energy alteration. Characteristics of macrophages after activation by media control or lipopolysaccharide with bacteria-free DNA (LPS + DNA) alone or with chloroquine (LPS + DNA + CQ) as indicated by extracellular acidification rate (ECAR) for a glycolysis stress test with the graph presentation of glycolysis capacity (maximal glycolysis) and glycolysis reserve (**A**–**C**), oxygen consumption rate (OCR) of mitochondrial stress test with the graph presentation of respiratory capacity (maximal respiration) and respiratory reserve (**D**–**F**), and mitochondrial abundance using mitochondrial DNA (*mtDNA*) and MitoTracker fluorescent staining (**G**,**H**) are demonstrated (independent triplicated experiments were performed). *, *p* < 0.05 vs. all of other groups; #, *p* < 0.05 vs. macrophages with media control group; the difference between groups was determined by one-way ANOVA with Tukey’s analysis.

**Figure 9 ijms-23-01907-f009:**
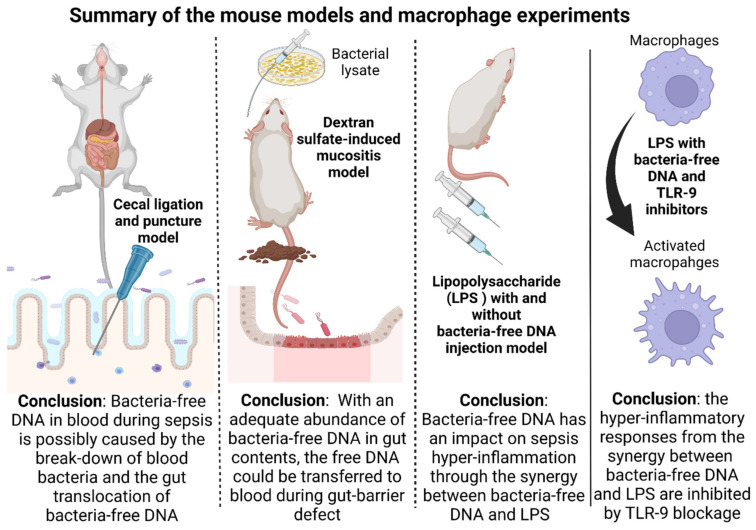
Conclusion of the experiments on mouse models and macrophages with the key massages (at the bottom part of the figure). Accordingly, (i) CLP (bacteria-free DNA possibly originated from the death of bacteria in blood and gut-barrier defect); (ii) DSS mucositis with oral gavage by bacterial lysate (bacteria-free DNA in blood was transferred from the gut); (iii) LPS with and without bacteria-free DNA injection (the importance of bacteria-free DNA in blood on the induction of LPS hyper-inflammatory responses); and (iv) the synergy between LPS and bacteria-free DNA on macrophage responses and the attenuation by TLR-9 blockage. Figure created using Biorender (https://biorender.com/, accessed on 29 January 2022).

**Figure 10 ijms-23-01907-f010:**
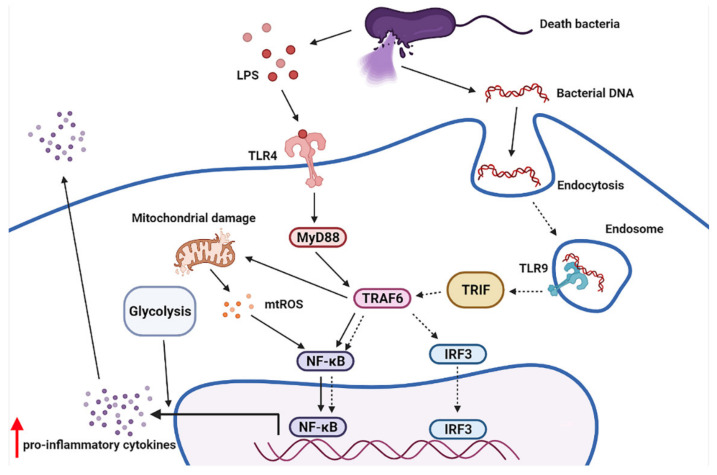
The working hypothesis demonstrates a simultaneous stimulation of surface TLR-4 and endosomal TLR-9 by LPS and bacteria-free DNA, respectively, which induces inflammatory responses through MyD88, and non-MyD88 (TRIF), which induces TRAF-6 [92,93]. The vigorous inflammatory stimulation, especially cytokine production, requires high cell energy, mostly by the glycolysis pathway [81], which might cause mitochondrial damage with higher inflammatory responses [75,94]. MyD88, myeloid differentiation primary response 88; TRIF, TIR-domain-containing adapter-inducing interferon-β; TRAF6, TNF receptor associated factor 6; IRF-3, interferon regulatory factor 3. Figure created using Biorender (https://biorender.com/, accessed on 29 January 2022).

**Table 1 ijms-23-01907-t001:** List of primers used in the study.

Primers	Forward	Reverse
Nuclear factor-kB (*NF-kB*)	5′-CTTCCTCAGCCATGGTACCTCT-3′	5′-CAAGTCTTCATCAGCATCAAACTG-3′
Toll like receptor-4 (*TLR-4*)	5′-GGCAGCAGGTGGAATTGTAT-3′	5′-AGGCCCCAGAGTTTTGTTCT-3′
Inducible nitric oxide synthase (*iNOS*)	5′-ACCCACATCTGGCAGAATGAG-3′	5′-AGCCATGACCTTTCGCATTAG-3′
Interleukin-1ß (*IL-1ß*)	5′-GAAATGCCACCTTTTGACAGTG-3′	5′-TGGATGCTCTCATCAGGACAG-3′
Arginase-1 (*Arg-1*)	5′-CTTGGCTTGCTTCGGAACTC-3′	5′-GGAGAAGGCGTTTGCTTAGTTC-3′
Transforming Growth Factor-β (*TGF-β*)	5′-CAGAGCTGCGCTTGCAGAG-3′	5′-GTCAGCAGCCGGTTACCAAG-3′
Resistin-like molecule-α (*FIZZ-1*)	5′-GCCAGGTCCTGGAACCTTTC-3′	5′-GGAGCAGGGAGATGCAGATGA-3′

## Data Availability

Data is contained within the article.

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
