# Peer review of "Blood Bacteria-Free DNA in Septic Mice Enhances LPS-Induced Inflammation in Mice through Macrophage Response"

_ijms, 2022, doi:10.3390/ijms23031907_

Round 1

Reviewer 1 Report

Thanks for reviewing and correcting the manuscript according to the reviewer's comments. Please, shorten the title and make it more clear and concise (in my opinion shortening the title this way is fine: "Blood bacteria-free-DNA in septic mice enhances LPS-induced inflammation in mice through macrophage response").

The abstract is confusing, please, re-write it only describing the main results and conclusions.

Author Response

Thanks for reviewing and correcting the manuscript according to the reviewer's comments. Please, shorten the title and make it clearer and more concise (in my opinion shortening the title this way is fine: "Blood bacteria-free-DNA in septic mice enhances LPS-induced inflammation in mice through macrophage response").

ANS: We thank the reviewer for the comment and use the provided title.

The abstract is confusing, please, re-write it only describing the main results and conclusions.

ANS: We thank the reviewer for the comment and re-write the abstract.

Reviewer 2 Report

The authors established that blood bacteria free DNA in sepsis mice may be transferred from the gut and led to exacerbation of LPS- induced inflammation through macrophage responses. The research is well designed and ethical. Manuscript is well written, discussion proper for this type of article. The research is multi-methodical and leads to interesting conclusions. The research, as the authors themselves have rightly pointed out, have a few limitations. It was not possible to prove gut translocation by direct intestinal imaging - which is the biggest limitation of the thesis, but it is highly probable. I have only minor comment to the manuscript. How many mice were used in each study? It is not clear. 

Author Response

The authors established that blood bacteria free DNA in sepsis mice may be transferred from the gut and led to exacerbation of LPS- induced inflammation through macrophage responses. The research is well designed and ethical. Manuscript is well written, discussion proper for this type of article. The research is multi-methodical and leads to interesting conclusions. The research, as the authors themselves have rightly pointed out, have a few limitations. It was not possible to prove gut translocation by direct intestinal imaging - which is the biggest limitation of the thesis, but it is highly probable. I have only minor comment to the manuscript. How many mice were used in each study? It is not clear. 

ANS: We thank the reviewer for the comment and add the information in method section.